# Shoot-to-root mobile CEPD-like 2 integrates shoot nitrogen status to systemically regulate nitrate uptake in *Arabidopsis*

Ryosuke Ota [1], Yuri Ohkubo [1], Yasuko Yamashita [1], Mari Ogawa-Ohnishi [1] & Yoshikatsu Matsubayashi [1]*

Plants modulate the efficiency of root nitrogen (N) acquisition in response to shoot N demand. However, molecular components directly involved in this shoot-to-root communication remain to be identified. Here, we show that phloem-mobile CEPD-like 2 (CEPDL2) polypeptide is upregulated in the leaf vasculature in response to decreased shoot N status and, after translocation to the roots, promotes high-affinity uptake and root-to-shoot transport of nitrate. Loss of *CEPDL2* leads to a reduction in shoot nitrate content and plant biomass. CEPDL2 contributes to N acquisition cooperatively with CEPD1 and CEPD2 which mediate root N status, and the complete loss of all three proteins severely impairs N homeostasis in plants. Reciprocal grafting analysis provides conclusive evidence that the shoot *CEPDL2/CEPD1/2* genotype defines the high-affinity nitrate uptake activity in root. Our results indicate that plants integrate shoot N status and root N status in leaves and systemically regulate the efficiency of root N acquisition.

[1] Division of Biological Science, Graduate School of Science, Nagoya University, Chikusa, Nagoya 464-8602, Japan. *email: matsu@bio.nagoya-u.ac.jp

Nitrogen (N) is an essential macronutrient that plays a crucial role throughout plant development. Most of the inorganic N in natural soils is present as nitrate as a result of microbial nitrification, but soil nitrate availability fluctuates temporally and spatially, depending on soil type and climatic conditions. To cope with such natural N environmental fluctuations, plants have evolved regulatory mechanisms that enable them to modulate the efficiency of root N acquisition in response to external N availability and their internal N status[1,2]. This systemic adaptive response involves mutual communication between distant tissues by means of long-distance signaling, as evidenced by the results of split-root experiments in which N deprivation in one part of the root system led to a compensatory increase in expression of the nitrate transporter gene NRT2.1 in the non-deprived part[2–4].

Recent molecular analyses have shown that a starvation-induced 15-amino-acid peptide hormone, CEP (C-terminally encoded peptide), acts as a root-derived ascending N-demand signal to the shoot via the xylem[5]. CEP is recognized in the shoot by the LRR-receptor kinase CEPR1 (CEP receptor 1), leading to the production of the non-secreted ≈100-amino-acid poly-peptides CEPD1 (CEP downstream 1) and CEPD2 as secondary signals[6]. CEPD1/2 act as shoot-derived descending signals to the root via the phloem and ultimately upregulate NRT2.1 expression in the non-deprived roots. Thus, local root N status, which is integrated by the shoots, can be reflected by the nitrate uptake efficiency of distant parts of the root system.

In addition to the N-demand signal originating from N-deprived roots, the plant hormone cytokinin (CK) is thought to communicate rhizosphere N availability to the shoots as an N-supply signal[7]. CK is synthesized in the roots in response to N supply and translocated to the shoots, leading to the promotion of shoot growth. In Arabidopsis, as loss of the CK transporter ABCG14—which mediates root-to-shoot translocation of trans-zeatin (tZ)[8,9] represses systemic N-demand signaling[10], integration of the tZ content as an N-supply signal in shoots may serve as a developmental checkpoint that ensures N acquisition by upregulating NRT2.1 expression in roots.

Despite significant progress in understanding the molecular mechanisms governing the integration of root N status, very little is known about how plants integrate shoot N status to systemically regulate N acquisition in roots. Shoot-derived phloem amino acids are often postulated as negative regulators of nitrate uptake, but there are not always positive correlations between the rate of nitrate uptake and the amino acid content of phloem[11]. Indeed, evidence from experiments with split-root systems has suggested that feedback regulation of nitrate uptake can occur independently of any change in the amino acid content of the phloem[12]. The actual shoot-derived signal that regulates root nitrate uptake in response to shoot N status remains to be identified. Here, we report the identification and functional characterization of this long-sought mobile signal. Our findings highlight how plants integrate internal nutritional cues and environmental conditions to optimize their growth and development.

## Results

### Shoot phloem-specific CEPD-like peptides upregulate NRT2.1.

We previously found that the N status of one part of the root system is communicated to other parts of the root system via root-to-shoot-to-root signaling by ascending CEP family peptides and descending CEPD1 and CEPD2 peptides[5,6]. The latter CEPD1/2 peptides belong to a large family of non-secreted peptides comprising 21 members in Arabidopsis (Supplementary Fig. 1a). Currently, these peptides are assigned to the plant-specific class III glutaredoxin family[13], but from a structural point

of view, there is controversy as to whether individual members function in redox regulation[14].

During the course of molecular analyses of this peptide family, we identified two additional peptides, At2g30540 and At3g62960, that upregulate the expression of NRT2.1 even under N-replete conditions (10 mM $NH_4^+$, 10 mM $NO_3^-$) when overexpressed (Fig. 1a). Both At2g30540 and At3g62960 are 102-amino-acid non-secreted polypeptides exhibiting high sequence similarity with CEPD1 (86% and 81% identity, respectively) (Supplementary Fig. 1b, c); therefore, we designated these peptides CEPD-like 1 (CEPDL1) and CEPDL2, respectively. Notably, the expression of NRT2.1 in CEPDL2-overexpressing plants (CEPDL2ox) increased by more than 25-fold relative to the wild-type (WT) control, a level far higher than that observed in CEPD1ox plants (Fig. 1a). Because NRT2.1 transcription typically is highly repressed in the presence of $NH_4^+$, the effect of CEPDL2-overexpression suggests a potentially important role for CEPDL2 in nitrate uptake pathway. We confirmed that the levels of CEPDL2 transcript were comparable among individual transgenic plants (Supplementary Fig. 1d). Under 10 mM $NO_3^-$ condition (devoid of $NH_4^+$) where basal NRT2.1 expression level is elevated, induction level of NRT2.1 in CEPDL2ox plants was 2.8-fold relative to the WT control grown under the same conditions (Fig. 1a).

GUS reporter-aided histochemical analyses revealed that the promoter activity of the CEPDL2 gene was prominent in the phloem cells in the cotyledons and mature leaves (Fig. 1b, c). qRT-PCR results confirmed that CEPDL2 transcripts accumulated to prominent levels in the shoots (Fig. 1d). A similar expression pattern was observed for CEPDL1 (Supplementary Fig. 1e–g). Based on the results of these molecular and histochemical analyses, we postulated that CEPDL peptides,

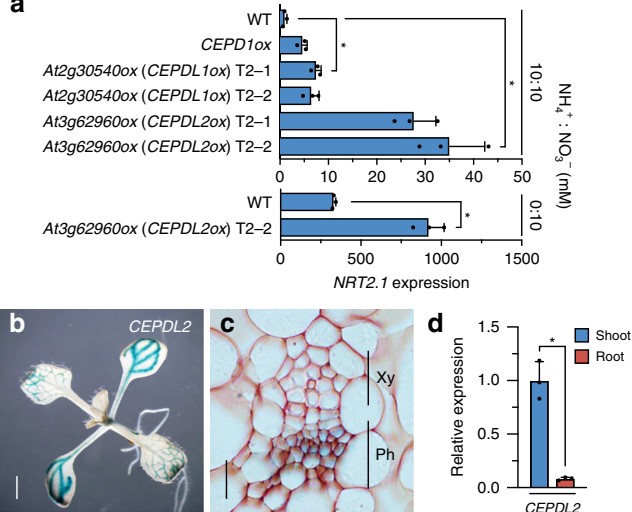

**Fig. 1 Overexpression of At3g62960 (CEPDL2) highly induces the expression of NRT2.1. a** NRT2.1 expression levels in the 10-day-old wild-type (WT) plants or transgenic plants overexpressing At2g30540 (CEPDL1) or At3g62960 (CEPDL2) under N-replete (10 mM $NH_4^+$, 10 mM $NO_3^-$) or 10 mM $NO_3^-$ conditions. For each transgene, results from two independent T2 transgenic lines are shown (mean ± SD, *P < 0.05 by two-tailed non-paired Student's t test, n = 3). Source data are provided as a Source Data file. **b** Histochemical staining of 10-day-old seedlings transformed with the CEPDL2pro:GUS gene. Scale bar = 1 mm. **c** Cross-section of the leaf vascular tissues pictured in **b**. Xy xylem, Ph phloem. Scale bar = 10 μm. **d** qRT-PCR analysis of CEPDL2 transcripts in the shoots and roots of 7-day-old WT plants cultured on 10 mM $NO_3^-$ condition (n = 3).

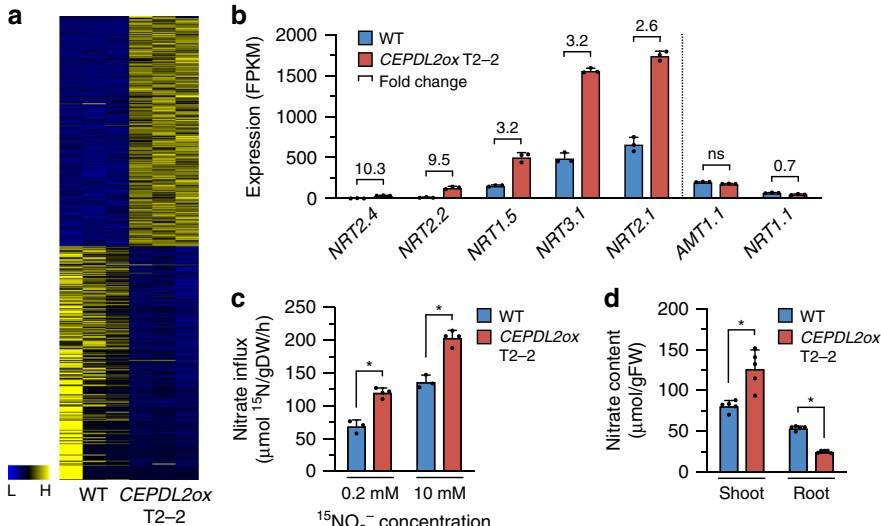

**Fig. 2 CEPDL2 regulates both high-affinity uptake and root-to-shoot transport of nitrate. a** Heatmap showing signal intensities of differentially expressed genes in three replicates each of 10-day-old WT and *CEPDL2ox* plant roots. **b** Nitrate transporter genes upregulated in roots of *CEPDL2ox* plants (*n* = 3). **c** Nitrate uptake activity of 10-day-old WT and *CEPDL2ox* plant roots, as determined by $^{15}NO_3^-$ influx at 0.2 mM and 10 mM (n = 4). **d** Nitrate content in shoots and roots of 16-day-old WT and *CEPDL2ox* plants (*n* = 5).

especially CEPDL2, play as yet uncharacterized roles in the systemic regulation, by the shoots, of root nitrate uptake.

**CEPDL2 regulates both uptake and transport of nitrate.** *CEPDL2ox* plants develop shorter roots and smaller rosettes compared to WT plants (Supplementary Fig. 2a–c). Genome-wide transcriptome analysis of roots of *CEPDL2ox* plants grown under 10 mM $NO_3^-$ conditions showed that 307 genes were upregulated and 311 downregulated as compared with WT roots (Fig. 2a, Supplementary Data 1). Gene ontology (GO) analysis of the top 150 upregulated genes (fold-change >2.5) showed highest enrichment in the molecular function "nitrate transporter activity" (Supplementary Fig. 2d). Indeed, the upregulated genes included the nitrate transporter genes such as *NRT2.4*, *NRT2.2*, *NRT3.1*, and *NRT2.1*, all of which are involved in high-affinity nitrate uptake[15–17], and *NRT1.5*, which loads nitrate into the xylem for root-to-shoot translocation[18] (Fig. 2b). By contrast, overexpression of *CEPDL2* had no positive effect on expression of the ammonium transporter gene *AMT1.1* or on that of the low-affinity nitrate transporter gene *NRT1.1*. The expression of the representative reference genes was stable in plants with and without *CEPDL2* overexpression (Supplementary Fig. 2e).

To verify further the function of CEPDL2, we measured root nitrate influx of *CEPDL2ox* plants at 0.2 and 10 mM $^{15}NO_3^-$ external concentrations to discriminate between the activities of the high- and low-affinity nitrate transport systems (HATS and LATS, respectively). The HATS provides a capacity for nitrate uptake at low (<0.5 mM) external nitrate concentration, whereas the LATS allows transport at high (>0.5 mM) external nitrate concentrations. When incubated in 0.2 mM $^{15}NO_3^-$, *CEPDL2ox* plants showed a 1.6-fold higher level of HATS activity than that in WT plants (Fig. 2c). In contrast, when LATS activity was calculated as the difference between root $^{15}NO_3^-$ influx measured at 10 and 0.2 mM, no significant difference was observed between the *CEPDL2ox* plants and WT plants, indicating that CEPDL2 preferentially regulates the high-affinity nitrate uptake system in roots (Supplementary Fig. 2f).

Analysis of tissue nitrate content in *CEPDL2ox* plants confirmed that shoot nitrate content was increased 1.4-fold compared to that of the WT (Fig. 2d). Importantly, the root nitrate content of

*CEPDL2ox* plants was only 47% of WT plants despite the elevated HATS activity in the roots. This result was interpreted as a consequence of increased loading of nitrates into the xylem following *NRT1.5* upregulation. Shoot nitrate reductase (NR) activity, which is rate limiting in the early steps of the nitrate assimilation pathway, was comparable in *CEPDL2ox* and WT plants (Supplementary Fig. 2g). These results strongly suggest that CEPDL2 acts in a pathway regulating both high-affinity uptake and root-to-shoot transport of nitrate.

**CEPDL2 expression is directly regulated by shoot N status.** Based on the assumption that CEPDL2 functions as a leaf-derived systemic signal that regulates root function, we examined how the expression of *CEPDL2* is regulated in terms of N homeostasis and environmental adaptation in plants. We previously showed that when roots are subjected to N starvation, *CEPD1/2* are upregulated in shoots in a CEPR1-dependent manner, indicating that descending CEPD1/2 signals exclusively reflect ascending CEP-mediated root N status[6]. From a similar point of view, we tested whether *CEPDL1* and *CEPDL2* are responsive to N deficiency in WT and *cepr1-1* mutant plants. When plants were cultivated on N-replete medium (10 mM $NH_4^+$, 10 mM $NO_3^-$) for 14 days and then transferred to N-depleted medium (0 mM $NH_4^+$, 0 mM $NO_3^-$) for 24 h, we found that expression of *CEPDL2*, but not that of *CEPDL1*, was markedly induced in shoots independent of CEPR1 (Supplementary Fig. 3a). This observation raised two possibilities: (i) *CEPDL2* responds to root N status–mediating signal(s) other than CEP family peptides, or (ii) *CEPDL2* is directly regulated by shoot N status.

To distinguish between the abovementioned possibilities, we examined whether *CEPDL2* is induced in detached shoots in which nutrients are directly applied through the cut hypocotyl. These conditions eliminated the influence of root-derived factors on leaf gene expression. We observed that when detached shoots experienced N starvation for 24 h, *CEPDL2* was highly induced in the leaves of both WT and *cepr1-1* plants (Fig. 3a). *CEPDL2* also was induced in the detached shoots of *cepr2-1* and *cepr1-1 cepr2-1* plants at levels comparable to those in WT (Supplementary Fig. 3b). *CEPDL2* expression in the detached shoots was repressed by supplementing the medium with nitrate, ammonium, or

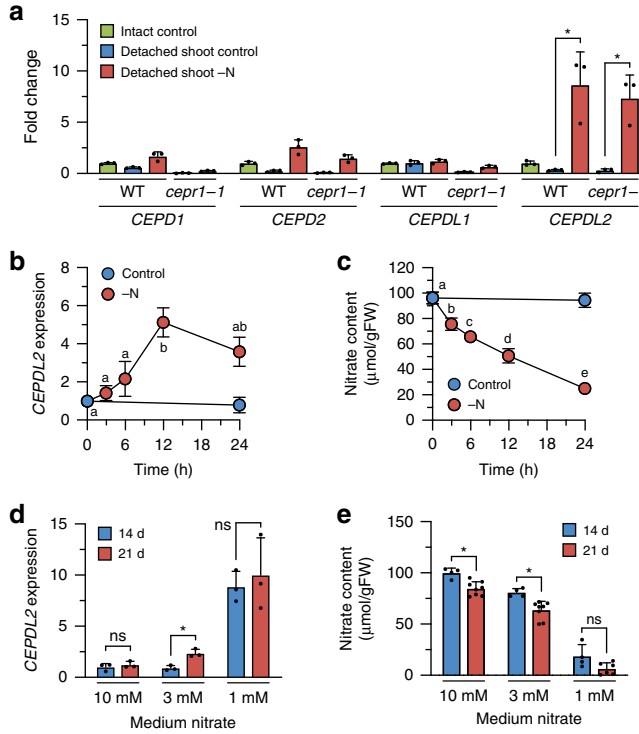

**Fig. 3 CEPDL2 expression is directly regulated by the N status of the shoots. a** qRT-PCR analysis of *CEPD1/2* and *CEPDL1/2* transcripts in the detached leaves of 12-day-old WT and *cepr1-1* plants after N starvation for 24 h (*n* = 3). **b** Changes in *CEPDL2* transcript levels in WT shoots subjected to N starvation for up to 24 h (*n* = 3). Different letters indicate statistically significant differences (*P* < 0.05, one-way ANOVA, *n* = 4). **c** Changes in nitrate content in WT shoots subjected to N starvation for up to 24 h (*n* = 4). **d** Levels of *CEPDL2* transcripts in WT shoots grown for 14 or 21 days under various nitrate conditions (*n* = 3). **e** Nitrate content in WT shoots grown under various nitrate conditions (*n* = 8).

glutamine at 10 mM, indicating that nitrate and its metabolites play a feedback role in regulating *CEPDL2* expression (Supplementary Fig. 3c).

The regulation of *CEPDL2* expression by shoot N status also was analyzed in terms of tissue nitrate content. When 14-day-old WT seedlings grown on $NO_3^-$-rich medium (10 mM $NO_3^-$) were transferred to N-deficient medium (0 mM $NO_3^-$), the level of *CEPDL2* expression increased between 6 and 12 h after the onset of N starvation, accompanied by a pronounced decrease in shoot nitrate content (Fig. 3b, c). In contrast, when WT seedlings that were pre-incubated under N-deficient conditions for 24 h were transferred to $NO_3^-$-rich medium, *CEPDL2* expression was downregulated 3 h after treatment, accompanied by the recovery of shoot nitrate content (Supplementary Fig. 3d, e). These results indicated that CEPDL2 expression is directly and reversibly regulated by the N status of the shoots.

In contrast to the case with *CEPDL2*, N starvation had no effect on *CEPDL1* expression, although *CEPDL1* overexpression led to *NRT2.1* upregulation in roots (Figs. 1a and 3a). As *CEPDL1* was nonresponsive to CEP[6], *CEPDL1* likely plays a role in regulating nitrate uptake but acts via a pathway other than that mediating the short-term response to N starvation.

We further analyzed the temporal expression patterns of *CEPDL2* in WT plants grown in the presence of various nitrate concentrations. Under $NO_3^-$-rich conditions (10 mM $NO_3^-$), expression of *CEPDL2* remained at the basal level at both 14 and 21 days after germination, timepoints that correspond to

high levels of nitrate content (>80 μmol/gFW) in plant shoots (Fig. 3d, e). The slightly decreased nitrate levels at 21 days relative to those in the 14-day plants could have been due to high N demand in the late vegetative stage (Fig. 3e). In contrast, when the medium $NO_3^-$ concentration was lowered to 3 mM, the *CEPDL2* transcript level increased 2.3-fold by 21 days. The shoot nitrate content decreased to 63.6 μmol/gFW under these conditions. Under $NO_3^-$-limited conditions (1 mM $NO_3^-$), shoot nitrate content decreased further at both 14 and 21 days (<20 μmol/gFW). In response to this low N status in shoots, *CEPDL2* was upregulated more than sixfold at both 14 and 21 days. These results indicated that the levels of *CEPDL2* expression are inversely proportional to the shoot nitrate content, which varies based on the $NO_3^-$ concentration in the medium and growth stage of the plants.

**Loss of *CEPDL2* impairs nitrate acquisition and plant growth.** Next, we investigated how the loss of CEPDL2 affects root nitrate uptake and plant growth. Because no T-DNA insertion mutant was available for the *CEPDL2* gene, we constructed a *cepdl2-1* deletion mutation using the CRISPR/Cas9 system and obtained two independent lines carrying the same 26-bp deletion that leads to a predicted frame shift (Supplementary Fig. 4a).

We cultured WT and *cepdl2-1* mutant plants on medium containing 10 mM, 3 mM or 1 mM $NO_3^-$ for 14 or 21 days, and then analyzed their phenotypes with respect to fresh weight and nitrate uptake activity. Under all $NO_3^-$ conditions, the *cepdl2-1* mutant displayed reduced shoot biomass at 21 days, accompanied by stunted leaves, symptoms that are similar to those resulting from N deficiency (Fig. 4a, b). On the culture medium containing 10 mM $NO_3^-$, a condition under which *CEPDL2* expression remained at the basal level (Fig. 3d), the fresh weight of *cepdl2-1* mutant was only 65.9% of that of WT plants at 21 days, suggesting that a basal level of CEPDL2 is needed for optimal shoot growth. On the medium containing 3 mM $NO_3^-$, a condition under which *CEPDL2* is upregulated at 21 days due to a decrease of nitrate content in the shoots (Fig. 3d, e), the fresh weight of *cepdl2-1* mutant plants was even lower, 57.5% of that of WT plants, indicating that CEPDL2 has a greater impact on shoot growth in the late vegetative phase under intermediate levels of rhizosphere N availability (Fig. 4b). This *cepdl2-1* phenotype was reproduced when plants were grown in vermiculite with nutrient solutions containing 3 mM $NO_3^-$ in the absence of sucrose (Supplementary Fig. 4b). When medium $NO_3^-$ was decreased to 1 mM, the fresh weight of *cepdl2-1* mutant plants was 76.0% of that of WT plants. Under $NO_3^-$-limiting conditions, the CEPD1/2 pathway is also upregulated in response to low N-status in the roots (Supplementary Fig. 4c), thus likely leading to a less pronounced phenotype in the *cepdl2-1* mutant.

Consistent with the growth phenotypes of the *cepdl2-1* mutant, HATS activity was significantly decreased in *cepdl2-1* mutant roots, with HATS activity falling to 65.8% of that of WT plants (Fig. 4c). Reduction in HATS activity is known to affect shoot growth under the millimolar $NO_3^-$ conditions as well as under micromolar range when grown in agar medium[19]. The root fresh weight of *cepdl2-1* mutant plants also was reduced compared to that of WT, whereas the primary root length was comparable between WT and *cepdl2-1* mutant plants (Supplementary Fig. 4d, e).

Complementation of the *cepdl2-1* mutant with *GFP-CEPDL2* reversed the shoot fresh weight and nitrate uptake activity to levels like those in the WT plants, confirming that these phenotypes are caused by loss of function of *CEPDL2* and that CEPDL2 is functional even when N-terminally fused to GFP (Supplementary Fig. 4f–h). The *cepdl2-1* plants also exhibited a reduction in both nitrate content and total N content in shoots at

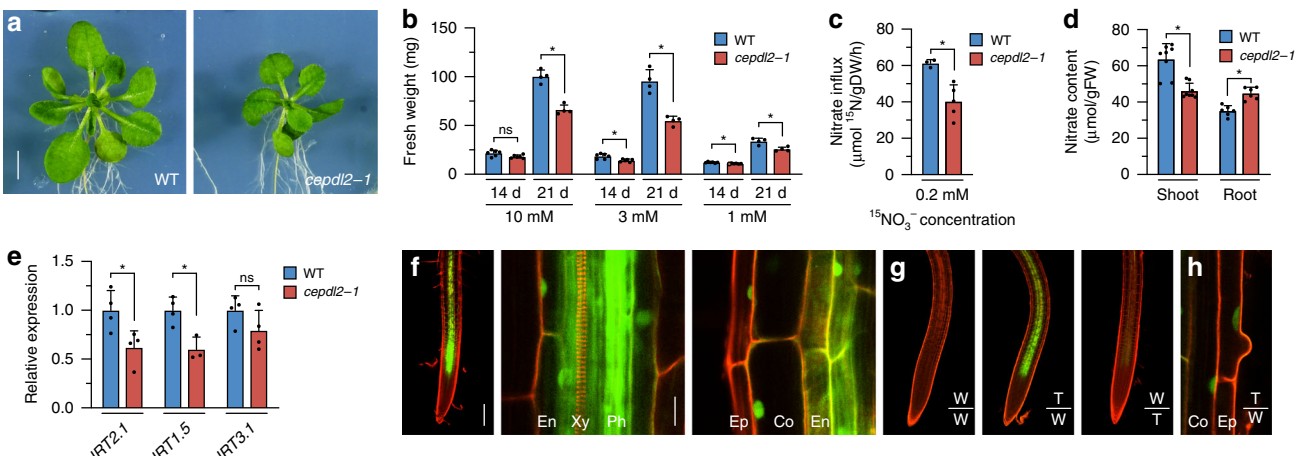

**Fig. 4 CEPDL2 functions as a shoot-to-root mobile signal and its loss results in impaired nitrate acquisition. a** Phenotypes of 21-day-old WT and *cepdl2-1* plants grown on $NO_3^-$-rich medium (10 mM $NO_3^-$). Scale bar = 5 mm. **b** Fresh weight of shoots of WT and *cepdl2-1* plants grown under various $NO_3^-$ conditions. Shoots were harvested at 14 or 21 days of culture (*n* = 6). **c** Reduction in HATS activity in 21-day-old *cepdl2-1* roots (*n* = 5). **d** Nitrate content in shoots and roots of 21-day-old WT and *cepdl2-1* plants (n = 6). **e** qRT-PCR analysis of *NRT2.1*, *NRT1.5* and *NRT3.1* transcripts in the roots of 21-day-old WT and *cepdl2-1* plants grown under 3 mM $NO_3^-$ condition (*n* = 4). **f** Detection of GFP-CEPDL2 signals in the root vascular region of 9-day-old WT plants expressing *GFP-CEPDL2* (left, scale bar = 200 μm) and its enlarged views (center and right, scale bar = 20 μm). Ep epidermis, Co cortex, En endodermis, Ph phloem, Xy xylem. **g** Detection of GFP-CEPDL2 signals in the root vascular region of reciprocally grafted plants. W wild-type plants, T transgenic plants expressing GFP-CEPDL2. **h** Enlarged view of roots of grafted plants in the T/W scion/rootstock combination.

21 days (Fig. 4d, Supplementary Fig. 4i). Importantly, the nitrate content in *cepdl2-1* roots was higher than that in WT roots (Fig. 4d). Combined with the observation that *cepdl2-1* mutant plants exhibited decreased expression of *NRT2.1* and *NRT1.5* (61.9% and 60.0% of the WT levels, respectively), we concluded that CEPDL2 positively regulates both high-affinity uptake and root-to-shoot transport of nitrate (Fig. 4e).

**CEPDL2 functions as a shoot-to-root mobile signal**. When a *GFP-CEPDL2* construct driven by the native *CEPDL2* promoter was introduced into WT plants, we observed GFP-CEPDL2 signals in the root vascular region (Fig. 4f). A closer view showed that GFP-CEPDL2 fluorescence localized predominantly in phloem cells but also in the endodermal, cortical, and epidermal cell layers. At the subcellular level, GFP-CEPDL2 localized in both the nucleus and cytoplasm in the endodermis, and notably, the fusion protein appeared to be more specifically confined to the nucleus in the cortex and epidermis (Fig. 4f). This nuclear localization pattern was specific to the fusion protein, differing from the pattern exhibited by the free-GFP control (Supplementary Fig. 4j).

We also performed reciprocal grafting between GFP-CEPDL2 and WT seedlings. When GFP-CEPDL2 scions were grafted onto WT rootstocks by hypocotyl-to-hypocotyl grafting, we observed accumulation of GFP-CEPDL2 signals in roots (Fig. 4g). No or only faint GFP signals were visible in roots when WT scions were grafted onto GFP-CEPDL2 rootstocks, in agreement with shoot-preferential *CEPDL2* expression (Fig. 1b–d). It should be noted that GFP-CEPDL2 was transported from the phloem to the surrounding endodermal, cortical, and epidermal cell layers even when GFP-CEPDL2 scions were grafted onto WT rootstocks (Fig. 4h). Collectively, these results indicate that the CEPDL2 polypeptide functions as a graft-transmissible phloem-mobile signal from the shoots to roots.

***t*Z-cytokinins are required for maximal induction of *CEPDL2*.** The *t*Z-type CK, which is synthesized in N-replete roots and

translocated to the shoots, is thought to communicate rhizosphere N availability to the shoots as an N-supply signal[7]. To determine if a decrease in *t*Z level in the shoots affects *CEPDL2* expression, we analyzed the levels of *CEPDL2* expression in an *abcg14* mutant defective in root-to-shoot translocation of *t*Z 24 h after transfer from N-replete to N-depleted medium. We observed that *CEPDL2* was induced even in the *abcg14* mutant after N deprivation, but the expression level was only 51% of that in the WT, suggesting that *t*Z-type CK is required for maximal induction of *CEPDL2* in shoots (Supplementary Fig. 4k). Importantly, loss of *ABCG14* caused a more drastic reduction in *CEPD1/2* levels after N deprivation, despite substantial upregulation of upstream CEP signaling, which explains the loss of systemic N-demand signaling in the *abcg14* mutant (Supplementary Fig. 4l)[10].

**CEPDL2 regulates N acquisition together with CEPD1/2**. To investigate further how the loss of CEPD1/2 and CEPDL2 polypeptides affects root nitrate uptake and plant growth, we cultured WT, *cepd1,2* double-, and *cepd1,2 cepdl2* triple-mutant plants on medium containing 10 mM, 3 mM or 1 mM $NO_3^-$ for 17 days (before bolting of multiple mutants). Under all $NO_3^-$ conditions, the *cepd1,2 cepdl2* triple mutant displayed severely reduced shoot biomass characterized by smaller rosettes compared with both WT and *cepd1,2* double mutant (Fig. 5a, Supplementary Fig. 5a). In medium with 3 mM $NO_3^-$, the fresh weight of *cepd1,2* double-mutant and *cepd1,2 cepdl2* triple-mutant shoots was 64.5 and 26.2% of that of WT plants, respectively. These double and triple mutants also exhibited a significant reduction in nitrate content in shoots (decreases of 52.4% and 12.6%, respectively), accompanied by an early-flowering phenotype at 21 days that is often induced by N deprivation (Supplementary Fig. 5b, c). In accordance with these shoot phenotypes of the *cepd1,2 cepdl2* triple mutant, high-affinity influx of $^{15}NO_3^-$ (HATS activity) in the triple-mutant root declined to only 53.2% of that of WT, accompanied by reductions in *NRT2.1* and *NRT3.1* gene expression (30.3 and 63.7% of WT, respectively) (Fig. 5b, c). Total N content of the triple-mutant shoot was 56.4% that of WT (Supplementary Fig. 5d). Complementation of the *cepd1,2 cepdl2*

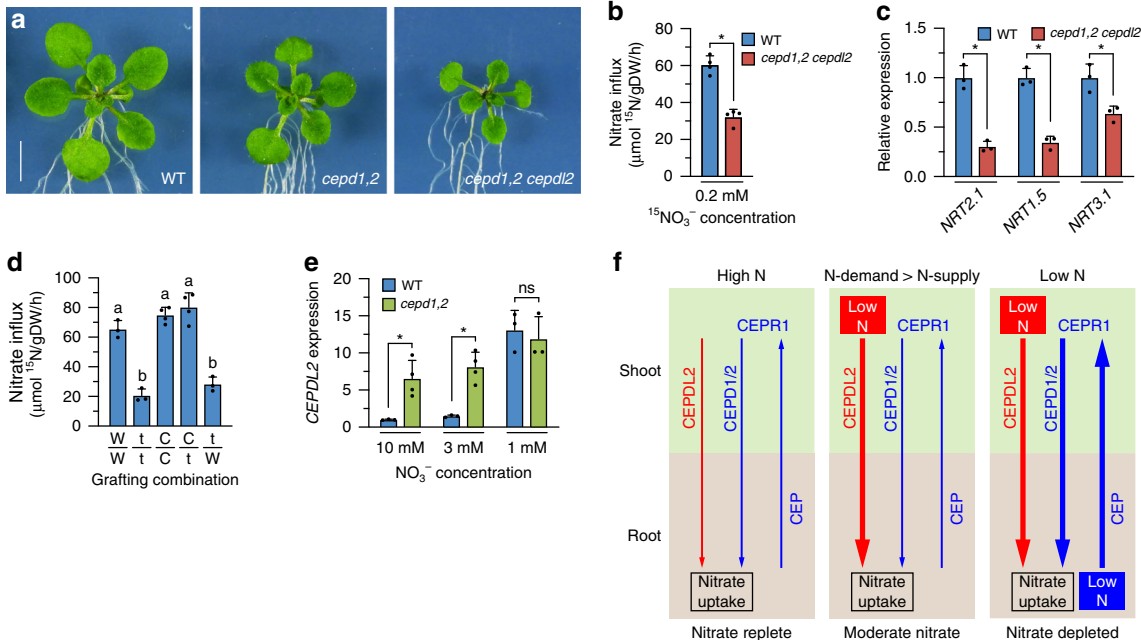

**Fig. 5 The CEPDL2 and CEPD1/2 act together from shoots to systemically regulate root nitrate acquisition. a** Phenotypes of 17-day-old WT and multiple-mutant plants grown on 3 mM $NO_3^-$ medium. Scale bar = 5 mm. **b** Reduction in HATS activity in *cepd1,2 cepdl2* triple-mutant roots ($n = 4$). **c** qRT-PCR analysis of *NRT2.1*, *NRT1.5* and *NRT3.1* transcripts in the roots of 17-day-old WT and *cepd1,2 cepdl2* triple-mutant plants ($n = 4$). **d** HATS activity in the root of reciprocally grafted plants. W wild type, t *cepd1,2 cepdl2* triple mutant, C *cepd1,2 cepdl2* triple mutant complemented with *GFP-CEPDL2* ($n = 3$). **e** qRT-PCR analysis of *CEPDL2* transcripts in the leaves of 17-day-old WT and *cepd1,2* mutant plants grown under 3 mM $NO_3^-$ condition ($n = 4$). **f** The emerging model of systemic regulation of nitrate acquisition by CEPDL2 and CEPD1/2 signaling. Thickness of arrows indicates strength of signal.

triple mutant with *GFP-CEPDL2* restored shoot fresh weight to the level obtained in the *cepd1,2* double-mutant plants (Supplementary Fig. 5e).

In contrast to these defects in root HATS activity and shoot nitrate content, we again observed that nitrate content and total N content in *cepd1,2 cepdl2* triple-mutant roots were comparable to those of WT plants (Supplementary Fig. 5b, d). This result is inferred to be a consequence of decreased loading of nitrate into the xylem by substantial downregulation of *NRT1.5* in the triple mutant (34.4% of WT) (Fig. 5c). Based on these collective results, we concluded that CEPD1/2 and CEPDL2 regulate both high-affinity uptake and root-to-shoot transport of nitrate.

The *cepd1/2 cepdl1/2* quadruple mutant displayed further reduced rosette development characterized by pale-green coloration of the leaves and early flowering (Supplementary Fig. 5a, f). Complementation with a *CEPDL1* genomic fragment restored shoot fresh weight to the level obtained in the *cepd1,2 cepdl2* triple mutant, confirming that this phenotype is caused by loss of function of *CEPDL1* (Supplementary Fig. 5g). Root HATS activity and shoot nitrate content of the quadruple mutant were only 23.9% and 3.0% of those of WT plants, respectively (Supplementary Fig. 5b, h). These results indicate that CEPDL1 also plays a regulatory role in N acquisition, a role that is redundant with those of CEPD1/2 and CEPDL2, even though *CEPDL1* transcript levels do not exhibit a short-term response to N starvation.

**Shoot *CEPDL2/CEPD1/2* genotype defines root HATS activity.** Reciprocal grafting analysis of WT, *cepd1,2 cepdl2* triple mutant and *cepd1,2 cepdl2* triple mutant complemented with *GFP-CEPDL2* provided further evidence that the shoot genotype defines the root HATS activity. When GFP-CEPDL2-complemented scions were grafted onto triple-mutant rootstocks, HATS activity of the root-stocks recovered to levels comparable to those of the parental complemented lines, even though the rootstocks themselves

expressed no *CEPDL2* (Fig. 5d). Shoot fresh weight was comparable to that of complemented plants (Supplementary Fig. 5i). In contrast, when *cepd1,2 cepdl2* triple-mutant scions were grafted onto WT rootstocks, HATS activity of the WT rootstocks decreased to the level of the triple mutant. These results indicate that CEPDL2 (and CEPD1/2) peptides expressed in the shoots are responsible for the systemic regulation of high-affinity nitrate uptake in roots.

**CEPDL2 acts in a compensatory manner to CEPD1/2.** We also examined whether CEPDL2 acts in a compensatory manner to CEPD1/2, which systemically regulate *NRT2.1* expression depending on root N status. Given that CEPDL2 is regulated by shoot N status, disruption of the CEPD1/2 pathway, which causes downregulation of nitrate uptake, should lead to compensatory upregulation of *CEPDL2* expression. To test this hypothesis, we analyzed the level of *CEPDL2* expression in shoots in the *cepd1,2* double mutant. When the mutant plants were cultured on 10 mM $NO_3^-$ medium for 17 days (before bolting), *CEPDL2* expression was upregulated 7.2-fold compared with that in WT plants, even though the medium nitrate concentration was high (Fig. 5e). Under 3 mM $NO_3^-$ conditions, in which basal *CEPDL2* expression in WT plants was elevated 1.5-fold compared with that in WT plants cultured on 10 mM $NO_3^-$, we observed that *CEPDL2* expression was upregulated 6.0-fold in the *cepd1,2* double mutant compared with that in WT plants grown under the same conditions. When medium $NO_3^-$ was further reduced to 1 mM, at which basal *CEPDL2* expression in WT plants was elevated more than 10-fold, no additional *CEPDL2* induction was detected in the *cepd1,2* mutant, suggesting saturation of the response. Collectively, these results indicate the existence of a dual system for regulating root nitrate uptake, in which the CEPD1/2 and CEPDL2 pathways function together to fine-tune expression of the nitrate transporters in response to fluctuations in the root N environment and shoot N status.

## Discussion

In conclusion, we propose that CEPDL2 is a shoot-derived descending signal that regulates both high-affinity uptake and root-to-shoot transport of nitrate in response to shoot N status. The emerging model of systemic regulation of nitrate acquisition by CEPDL2 and CEPD1/2 is as follows (Fig. 5f). (I) When roots are grown in a nitrate-replete environment and take up sufficient N to meet shoot N demand, the activity of both the CEPDL2 and CEPD1/2 pathways is maintained at basal levels. Note that because shoot fresh weight of the cepd1,2 cepdl2 triple mutant was as low as 21.6% of that in WT plants in the presence of 10 mM $NO_3^-$ (Supplementary Fig. 5a), basal level activation of these pathways is critical for nitrate acquisition even in nitrate-replete environments. (II) Even when plants are grown under nitrate-replete or -moderate conditions, CEPDL2 is upregulated when the root N uptake is insufficient to meet the N demand of shoots. This compensatory upregulation of CEPDL2 was prominent in the cepd1,2 mutant (Fig. 5e) and in WT plants grown at 3 mM $NO_3^-$ in the late vegetative phase (21 days), a phase in which shoot N demand is high (Fig. 3d). Indeed, the reduction in shoot fresh weight of the cepdl2-1 mutant was most pronounced at 21 days on medium containing 3 mM $NO_3^-$ (Fig. 4b). (III) When roots are subjected to nitrate-limited conditions and cannot take up sufficient nitrate to meet shoot N demand, both the CEPDL2 and CEPD1/2 pathways are highly upregulated (Fig. 3d, Supplementary Fig. 4c). As plants adapt to the low N conditions by reducing the shoot size (Fig. 4b), the net output of descending signals produced in the leaf phloem cells varies as a function of gene expression and corresponding leaf area. The magnitude of CEPDL2 and CEPD1/2 signaling appears to be tightly balanced between environmental N availability, the plant's own N status, and the shoot/root ratio.

CEPDL2 expression in the detached shoots was repressed by supplementing the medium with nitrate, ammonium, or glutamine, indicating that nitrate and its metabolites play a feedback role in regulating CEPDL2 expression (Supplementary Fig. 3c). However, there is still an open question as to how plants sense their N status[11]. Several lines of evidence suggest that candidates of the N sensory systems include pathways that have been described in yeast or other organisms such as the target of rapamycin (TOR) pathway[20], the plastidic PII-dependent pathway[21], and the family of glutamate receptor-like (GLR) proteins[22]. However, despite progress in elucidating the function and mode of action of these signaling systems, there is still much uncertainty about the extent to which these systems contribute to how plants monitor their N status[11]. The mechanism for inducing CEPDL2 in response to shoot N status remains an intriguing question that will need to be addressed in future work.

We also found that tZ-type CK, the predominant form of CK in xylem sap, is required in shoots for maximal induction of CEPDL2 and CEPD1/2 expression (Supplementary Fig. 4k). tZ-type CK is synthesized in roots in response to N supplementation, and thus, it is thought to communicate rhizosphere N availability to the shoots as an N-supply signal[7]. Considering the cost-benefit tradeoffs associated with the N-acquisition system, it is plausible that tZ-mediated N-supply signaling is a prerequisite for upregulation of CEPDL2 and CEPD1/2 signaling that ultimately promotes nitrate acquisition. Taken together, our results depict a sophisticated regulatory system in which the mature leaves of shoots integrate shoot N status, root N status, and rhizosphere N availability and compute the optimal level of root N acquisition to adapt to natural fluctuations in N availability. Given the relatively simple structural organization of plant bodies, it is surprising that such a convergent long-distance signaling mechanism evolved.

## Methods

**Plant materials and growth conditions**. Arabidopsis thaliana ecotype Nössen was used as WT, except for overexpression analysis of CEPDLs, in which Col ecotype was used as WT. The cepr1-1, cepr2-1, cepr1-1 cepr2-1, and cepd1-1 cepd2-1 mutants (Nössen background) were described previously[5,6]. The cepdl1-1 mutant was obtained from the RIKEN Arabidopsis transposon mutant collection (pst10389, Nössen background)[23]. The cepdl2-1 mutant was generated using a CRISPR/Cas9 system. The gRNA sequence ATGAGAATGTCATCGGAGAA was cloned into pKIR1.0[24], and directly transformed into the cepd1-1 cepd2-1 double mutant. We screened 32 lines and found two with the same 26-bp deletion in the coding region of CEPDL2. The CRISPR/Cas9 construct then was removed to ensure genetic stability. The cepdl2-1 single mutant was obtained by backcrossing the cepd1-1 cepd2-1 cepdl2-1 triple mutant to WT Nössen. The Arabidopsis abcg14 mutant was a kind gift from T. Kiba (Nagoya University)[9]. Surface-sterilized seeds were sown in plates on modified Murashige-Skoog (MS) solid medium. Modified MS medium for N-rich conditions (N-replete medium) contains 10 mM $NH_4Cl$ and 10 mM $KNO_3$ as the N and K sources, respectively, and half-strength concentrations of the other elements and 0.5% sucrose, and was adjusted to pH 5.7 with KOH. For 10 mM $NO_3^-$ medium, 10 mM $KNO_3$ was added solely as the N and K sources. For 3 mM $NO_3^-$ medium, 3 mM $KNO_3$ and 7 mM KCl were added. For 1 mM $NO_3^-$ medium, 1 mM $KNO_3$ and 9 mM KCl were added. For N-depleted medium, 10 mM KCl was added solely as the K source. All the plants were grown on vertically-oriented agar plates. For vertical culture, 12 seeds were sown on medium solidified using 1.5% agar in 13 × 10 cm plastic plates. Plants were grown at 22 °C with continuous light at an intensity of 80 μmol·m$^{-2}$·s$^{-1}$. At least three independent biological replicates were performed for each experiment.

**N-starvation assay**. N-starvation treatment of the plants was conducted by transferring the 14-day-old seedlings, grown vertically on N-replete agar medium (10 mM $NH_4^+$, 10 mM $NO_3^-$) or 10 mM $NO_3^-$ agar medium, to N-depleted agar medium. For the N-starvation treatment of the detached shoots, shoots of 12-day-old seedlings grown on N-replete agar medium were cut below the cotyledons and transferred to N-depleted liquid medium or N-replete liquid medium as control.

**Overexpression analysis**. We cloned the CEPDL1 or CEPDL2 ORFs into XbaI-, SacI-digested pBI121 vector downstream of the CaMV 35 S promoter using the Gibson Assembly system (New England Biolabs). General sequence analysis was performed using GENETYX-MAC software (Genetyx, Tokyo, Japan). Transgenic Arabidopsis plants were generated by the standard Agrobacterium-mediated transformation using floral dip method. The primer list is shown in Supplementary Data 2.

**RNA-seq analysis**. Total RNA (1 μg) was used to prepare RNA-seq libraries, using a TruSeq RNA Sample Prep Kit v2 (illumina) according to the manufacturer's instructions. Multiplexed libraries were sequenced on an illumina MiniSeq with single-end 75-bp sequencing. RNA-seq data were mapped to the Arabidopsis TAIR10 genome release using the bioinformatic pipeline of the illumina Base Space Sequence Hub. Genes with a q value < 0.05 and absolute log$_2$ fold change > 1 were defined as differentially expressed genes. GO enrichment analysis was performed using the PANTHER database.

**Real-time qPCR**. Total RNA was prepared from roots or shoots of the plants grown in the indicated conditions using an RNeasy kit (Qiagen). First-strand cDNA was synthesized from 2 μg of root-derived total RNA using the Superscript IV first strand synthesis system (Invitrogen) according to the manufacturer's protocol. Primers and probes were designed using Probe Finder software in the Universal Probe Library (UPL) assay design center (Roche Applied Science, Germany). Real-time qPCR was performed using a StepOne System (Applied Biosystems). Constitutively expressed EF-1α was used as a reference gene for normalization of the qRT-PCR data. List of the primer and UPL probe is provided in Supplementary Data 2. All statistical analysis was performed by using GraphPad Prism version 8 (GraphPad software).

**Promoter GUS analysis**. For β-glucuronidase (GUS) reporter-aided analysis of the promoter activities of CEPDL1 and CEPDL2, we amplified the 2.0-kb upstream sequences of the predicted ATG start codons of each gene by genomic PCR, and cloned the fragments into a promoter-less pBI101 vector upstream of the GUS reporter gene using the In-Fusion cloning system (Clontech). GUS activity was visualized using X-Gluc as substrate using a conventional protocol.

**GFP-CEPDL2 expression**. For expression analysis of GFP-CEPDL2, 2.0-kb upstream sequences from the predicted ATG start codon of CEPDL2, the GFP-coding region and the CEPDL2 ORF were ligated in-frame in this order into the HindIII-, BamHI-digested binary vector pCAMBIA1300-BASTA by four-component ligation using the In-Fusion cloning system. The resulting GFP-CEPDL2 constructs were introduced into the WT plants or the cepdl2-1 single and cepd1-1 cepd2-1 cepdl2-1 triple mutants (for complementation test).

**Imaging and microscopy**. For leaf sectioning, leaves were fixed in FAA solution (3.7% formaldehyde, 5% acetic acid, and 50% ethanol), dehydrated through a graded ethanol series, and embedded in Technovit 7100 resin (Heraeus Kulzer, Germany) following the manufacturer's protocol. Sections were cut at 5-μm thicknesses using a rotary microtome (Leica RM2235), counter-stained with 0.05% Nile red, mounted with Entellan (Merck) and observed under a standard light microscope (Olympus BX60). For root imaging, cell outlines were stained with 50 μg/ml propidium iodide for 2 min and observed under a confocal laser-scanning microscope (Olympus FV300) with helium-neon laser excitation at 543 nm. GFP images were collected with argon laser excitation at 488 nm.

**Grafting experiments**. *Arabidopsis* seedlings were grown vertically for 8 days on 10 mM $NO_3^-$ agar medium before grafting at 22 °C with continuous weak light at an intensity of 40 μmol·m$^{-2}$·s$^{-1}$ to allow hypocotyl elongation. Grafting was performed aseptically by cutting the rootstock donor perpendicular to the hypocotyl using a surgical blade, then inserting the rootstock into a short length (≈2 mm) of sterile 0.4-mm-diameter silicon tubing as described previously[25] with minor modifications. The scion was excised in a similar manner and inserted into the other end of the tubing until the scion touched the rootstock. Grafted plants were grown with continuous light at an intensity of 80 μmol·m$^{-2}$·s$^{-1}$ on 3 mM $NO_3^-$ agar medium for 10–20 days, until root length reached 9 cm.

**Nitrogen content and root $^{15}$N influx**. Seedlings were sequentially transferred to 0.1 mM $CaSO_4$ for 1 min and to basal liquid medium containing 0.2 mM or 10 mM $^{15}NO_3^-$ as N source for 10 min. At the end of the $^{15}$N labeling, roots were washed for 1 min in 0.1 mM $CaSO_4$ and were separated from shoots. The roots were lyophilized *in vacuo* and analyzed for total N and $^{15}$N contents by elemental analysis–isotope ratio mass spectrometry (Flash EA1112-DELTA V PLUS ConFlo III System, Thermo Fisher Scientific). For analysis of nitrogen content of shoots, lyophilized shoots were subjected directly to elemental analysis.

**Nitrate content**. The $NO_3^-$ ion concentration in the tissue was measured by an ion chromatography system (Dionex Aquion, Thermo Fisher Scientific). Shoots or roots were powdered in liquid nitrogen and mixed with 1 ml of water to extract nitrate. After centrifugation, the crude tissue extract was diluted tenfold with water and 25-μl aliquots were analyzed on a Dionex IonPac AS22 column (4 mm i.d. · 250 mm) for 15 min. The mobile phase consisted of an eluent containing 4.5 mM $Na_2CO_3$ and 1.4 mM $NaHCO_3$, employed at a flow rate of 1.2 ml/min at 30 °C with a conductivity detector equipped with Dionex AERS 500 suppressor unit.

**Nitrate reductase activity**. Nitrate reductase activity was determined as described[26] with modifications. Briefly, shoots of 17-day-old seedlings were powdered in liquid nitrogen and mixed with extraction buffer at 0.5 mg fresh weight (FW)/μl. After centrifugation, 15 μl of the crude extract was combined with the 85 μl of reaction buffer, and the mixture was incubated at 25 °C for 2 h. To quantify the amount of generated nitrite, 25 μl of the reaction mixture was analyzed by an ion chromatography system (Dionex Aquion, Thermo Fisher Scientific) on the Dionex IonPac AS22 column (4 mm i.d. · 250 mm) for 15 min. The mobile phase consisted of an eluent containing 4.5 mM $Na_2CO_3$ and 1.4 mM $NaHCO_3$, employed at a flow rate of 1.2 mL/min at 30 °C.

**Reporting summary**. Further information on research design is available in the Nature Research Reporting Summary linked to this article.

## Data availability
The RNA-seq data have been deposited in the NCBI Gene Expression Omnibus under accession number GSE140834 [https://www.ncbi.nlm.nih.gov/geo/query/acc.cgi?acc=GSE140834]. The raw FASTQ files have been deposited in the NCBI Sequence Read Archive under accession number SRP231423 [https://trace.ncbi.nlm.nih.gov/Traces/sra/?study=SRP231423]. The source data underlying Figs. 1–5 and Supplementary Figs. 1–5 are provided as a Source Data file. The *Arabidopsis* lines generated in this study are available from the corresponding author on reasonable request.

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

## Acknowledgements
This research was supported by a Grant-in-Aid for Scientific Research (S) (No. 18H05274) and a Grant-in-Aid for Scientific Research on Innovative Areas (No. 15H05957) from the Japan Society for the Promotion of Science.

## Author contributions
Y.M. conceived this project and designed the experiments with input from all co-authors. R.O., Y.O., Y.Y., M.O., and Y.M. performed the experiments and interpreted the results. Y.M. wrote the manuscript with input from all co-authors.

## Competing interests
The authors declare no competing interests.
