## [Peer Review File · Nature Communications]

Reviewers' comments:

Reviewer #1 (Remarks to the Author):

This manuscript by Ota and colleagues presents an important piece in the puzzle of how plants modulate root nitrogen uptake in response to the amount of nitrogen available in the shoot. Previously, the group of Matsubayashi have shown that CEP peptides (CEP1) are upregulated in the stele of lateral roots in response to N starvation and are loaded into xylem vessels where they are transported to the shoot. Here they relay a signal through two related receptors CEPR1 and 2 expressed in the vascular tissue of leaves, leading to the production of a shoot signal that ultimately upregulates nitrate transporter genes in the roots (Science 2014). They later showed that the phloem-specific signal that was induced in leaves upon perception of the root-derived CEPs were the CEP DOWNSTREAM 1 (CEPD1) and CEPD2 polypeptides (Nature Plants 2017). Here Ota and colleagues identify yet another phloem-mobile polypeptide CEPDL2 that is upregulated in the leaf vasculature in response to decreased shoot nitrogen availability and is transported to the root to activate the expression of genes encoding nitrate transporters. Unlike the CEPD1/2 polypeptides the CEPDL2 peptide is upregulated independent of the presence of the CEPs (shown by *cepr1* mutant studies). The authors provide a model where the CEPDL2 and CEPD1/2 pathways act complementary to regulate nitrogen in the plant as a response to environmental availability, the amount present in the plant and the shoot to root ratio of nitrogen.

The manuscript is well structured, clearly and precisely written. The results and figures are of high quality and support the conclusions drawn by the authors. The statistical analysis is satisfactory and has been applied where needed, even if I would have liked an explanation for the letters in figure 1A and elsewhere in the supplementary material. The discussion appropriately places the work presented in a context of previously published work relevant for the results presented here. I evaluate this work as solid and interesting but I am not completely certain that it is novel enough to merit publication in Nature Communications. As such the authors clearly present a new player in nitrogen homeostasis in plants but mechanistically it is not very different from previous publications.

The authors show in figure 3 that the regulation of CEPDL2 is independent of the CEPs and directly regulated by the nitrogen status in the shoot by measuring the relative expression of CEPDL2 in detached leaves subjected to nitrogen depletion. They use leaves from WT and the *cepr1* mutant and show that CEPDL2 is induced in nitrogen depleted leaves in both genotypes compared to intact and detached controls. In my opinion since the authors have previously shown that CEPR2 is also capable of binding CEP peptides and relaying a cellular signal from these peptides, these experiments should also be done on *cepr2* single and *cepr1 cepr2* double mutants. Alternatively, one could investigate the expression level of CEPR2 in *cep* mutant(s). The authors should try to address which signals (if not the CEPs) in the shoot are responsible for the regulation of CEPDL2.

Minor comments:

Line 48-52: Please rephrase it is difficult to understand this sentence.

Line 89: I think there is a reference to supplementary figure S1 C missing here.

Lines 109-114: Could the HATS and LATS be explained better. I am not sure it is obvious what is meant by LATS activity.

Reviewer #2 (Remarks to the Author):

The manuscript adds novelty to the previously reported phloem mobile peptide signalling model. Novel, but closely relatives of previously reported peptides are identified to also have a N

signalling role.

TITLE: The title should include the new peptides name to make clear the novelty to readers. However, naming the CEPD polypeptides is confusing for readers. Why call it 'CEPD-like'? Can these polypeptides be other member of the same family? It is confusing to use the tag on the name 'like'. (See alignment in Fig.S1 they are very similar).

The structural relationships between the CEPD-like peptides and the previously reported peptide signals CEP/CEPD needs to be made clear to the reader perhaps in a family tree.

Do the CEPD-like peptides interact with the LRR-receptor kinase CEPR1 (CEP receptor 1)?

Page 13. Line 389. Only one reference gene is used for the real-time qPCR, usually two are used. What checks were done to check how reference gene expression (EF-1 α) changes in transgenic plants?

(See Gutierrez L, et al. (2008) The lack of a systematic validation of reference genes: a serious pitfall undervalued in reverse transcription-polymerase chain reaction (RT-PCR) analysis in plants. *Plant Biotechnology Journal* 6 (6):609-618.)

Figure 2. The CEPDL2ox line shows much greater (two-fold more) nitrate influx and nitrate content – yet the plants grow slower than WT (see Fig S2). Nitrate uptake and supply is usually regarded as a driver for growth so how do the authors explain this result? This result suggests the CEPDL2ox line has uptake that is not linked to assimilation. Was nitrate reductase activity measured?

Furthermore, these differences in growth rate may be very important for the chose of reference genes for the expression analysis. The data in Figure S2 shows the CEPDL2ox line is significantly smaller.

Was only one CEPDL2ox line generated? The data seems to suggest there is only one line, therefore the phenotype should be checked in several ox lines.

Figure S1. (A) The CEPD1/2 and CEPDL1/2 peptide sequences alignments show they are very similar. Are the PCR primers able to distinguish adequately between expression of these very similar genes? The authors should show DNA sequence alignments too and their primer design.

Line 57. The reference is 17 years old – are there only a few correlations between nitrate uptake rate and amino acid content of the phloem? There are several papers reporting links between phloem amino acids and nitrate uptake.

Lines 59-60. The authors have previously shown mobile peptide signals – why not mention their own previous work on CEP/CEPD signals at this point in the text?

Response to referees

Response to Reviewer #1:

1) *Nitrogen depletion experiments should also be done on cepr2 single and cepr1 cepr2 double mutants.*

We confirmed that *CEPDL2* also was induced in the detached shoots of *cepr2-1* single and *cepr1-1 cepr2-1* double-mutant plants at levels comparable to those in WT (Supplementary Fig. 3b). These results further support the conclusion that *CEPDL2* expression is directly regulated by the N status of the shoots.

2) *The authors should try to address which signals (if not the CEPs) in the shoot are responsible for the regulation of CEPDL2.*

CEPDL2 expression in the detached shoots was repressed by supplementing the medium with nitrate, ammonium, or glutamine at 10 mM, indicating that nitrate and its metabolites play a feedback role in regulating *CEPDL2* expression (Supplementary Fig. 3c). However, it is still an open question how plants sense their N status in shoots. A recent review by Forde and colleagues (*J Exp Bot* **68**, 2531-2540 (2017)) notes that “candidates that are considered for the role of N sensory systems include the target of rapamycin (TOR) signaling pathway, the general control non-derepressible 2 (GCN2) pathway, the plastidic PII-dependent pathway, and the family of glutamate receptor-like (GLR) proteins. However, despite significant recent progress in elucidating the function and mode of action of these signaling systems, there is still much uncertainty about the extent to which they contribute to the process by which plants monitor their N status”. This is one of the important questions in this field to be solved in the future. We have included this point in the revised Discussion section.

3) *Line 48-52: Please rephrase it is difficult to understand this sentence.*

This sentence in the introduction was rephrased to “Integration of the *tZ* content as an N-supply signal in shoots may serve as a developmental checkpoint that ensures the N acquisition by upregulating *NRT2.1* in roots”.

4) *Line 89: I think there is a reference to supplementary figure S1 C missing here.*

We added a reference to that figure in this sentence.

5) *Lines 109-114: Could the HATS and LATS be explained better.*

We added the following explanatory sentence here: “The HATS provides a capacity for nitrate uptake at low (< 0.5 mM) external nitrate concentration, whereas the LATS allows transport at high (> 0.5 mM) external nitrate concentrations”.

Response to Reviewer #2:

1) The title should include the new peptides name to make clear the novelty to readers.

According to this suggestion, we changed the title to “Shoot-to-root mobile CEPD-like 2 integrates shoot nitrogen status to systemically regulate nitrate uptake in *Arabidopsis*”.

2) The structural relationships between the CEPD-like peptides and the previously reported peptide signals CEPD needs to be made clear to the reader perhaps in a family tree.

As suggested, we added a phylogenetic tree of the class III glutaredoxin family including CEPDs and CEPDLs (Supplementary Fig. 1a).

3) Do the CEPD-like peptides interact with the CEPR1 (CEP receptor 1)?

We currently are studying the endogenous interactors of CEPDs/CEPDLs by immunoprecipitation mass spectrometry. To date, we have not detected co-immunoprecipitation of CEPR1 with GFP-CEPDL2. However, these preliminary data are not included in the submitted manuscript.

*4) What checks were done to check how reference gene expression (*EF-1 α*) changes in transgenic plants?*

We used transcriptome analysis to confirm that the expression of representative reference including *EF-1 α* , was stable even in the transgenic plants (Supplementary Fig. 2e).

5) Was nitrate reductase activity in the CEPDL2ox line measured?

We measured shoot activity of nitrate reductase (NR), which is a rate-limiting reaction in the early steps of the nitrate assimilation pathway. Results show that NR activity was comparable between *CEPDL2ox* and WT plants (Supplementary Fig. 2g).

6) CEPDL2ox phenotype should be checked in several ox lines

For each transgene, results from two independent T2 transgenic lines are shown in the revised

manuscript (Fig. 1, Supplementary Fig. 1, Supplementary Fig. 2).

7) For the CEPD1/2 and CEPDL1/2 peptide sequences, the authors should show DNA sequence alignments too and their primer design.

We provided DNA sequence alignments in Supplementary Fig. 1c and the primer and Universal Probe Library (UPL) probe design in Supplementary Table 2. UPL probes are designed to hybridize gene-specific regions of each gene.

8) The reference is 17 years old. There are several papers reporting links between phloem amino acids and nitrate uptake.

We replaced the reference with a more recent review article (Gent & Forde, 2017). This review article notes that “shoot-derived phloem amino acids are often postulated as negative regulators of nitrate uptake, but there are not always positive correlations between the rate of nitrate uptake and the amino acid content of phloem”. Indeed, evidence from experiments with split-root systems have suggested that feedback regulation of nitrate uptake can occur independently of any change in the amino acid content of the phloem (Tillard *et al. J Exp Bot* **49**, 1371-1379 (1998)).

9) The authors have previously shown mobile peptide signals – why not mention their own previous work on CEPD signals at this point in the text?

We mentioned the CEP-CEPR-CEPD system that mediates root N status in the second paragraph of the Introduction. However, because the main subject of the last paragraph of the Introduction is how plants integrate shoot N status to systemically regulate N acquisition in the roots, we chose not to mention CEPD signals again at this point.

REVIEWERS' COMMENTS:

Reviewer #1 (Remarks to the Author):

I have now revised the manuscript by Ota and colleagues and am pleased to see that they have addressed my previous concerns. I am satisfied with this revision and would be glad to see this work published in Nature Communications. I am still curious to know which signals in the shoot are responsible for the regulation of CEPDL2 and hope that this is a question the authors address in their future line of research.

Reviewer #2 (Remarks to the Author):

The authors have fully addressed all the points I raised.